# Memory-Efficient Fine-Tuning of Compressed Large Language Models via sub-4-bit Integer Quantization

**Jeonghoon Kim**[*]
NAVER Cloud
jeonghoon.samuel@gmail.com

**Jung Hyun Lee**[*]
NAVER Cloud
onliwad101@gmail.com

**Sungdong Kim**
NAVER Cloud, KAIST AI
sungdong.kim@navercorp.com

**Joonsuk Park**
NAVER Cloud, NAVER AI Lab,
University of Richmond
park@joonsuk.org

**Kang Min Yoo**
NAVER Cloud, SNU AI Center
kangmin.yoo@gmail.com

**Se Jung Kwon**
NAVER Cloud
sejung.kwon@navercorp.com

**Dongsoo Lee**
NAVER Cloud
dongsoo.lee@navercorp.com

## Abstract

Large language models (LLMs) face the challenges in fine-tuning and deployment due to their high memory demands and computational costs. While parameter-efficient fine-tuning (PEFT) methods aim to reduce the memory usage of the optimizer state during fine-tuning, the inherent size of pre-trained LLM weights continues to be a pressing concern. Even though quantization techniques are widely proposed to ease memory demands and accelerate LLM inference, most of these techniques are geared towards the deployment phase. To bridge this gap, this paper presents Parameter-Efficient and Quantization-aware Adaptation (PEQA) – a simple yet effective method that combines the advantages of PEFT with quantized LLMs. By updating solely the quantization scales, PEQA can be directly applied to quantized LLMs, ensuring seamless task transitions. Parallel to existing PEFT methods, PEQA significantly reduces the memory overhead associated with the optimizer state. Furthermore, it leverages the advantages of quantization to substantially reduce model sizes. Even after fine-tuning, the quantization structure of a PEQA-tuned LLM remains intact, allowing for accelerated inference on the deployment stage. We employ PEQA-tuning for task-specific adaptation on LLMs with up to 65 billion parameters. To assess the logical reasoning and language comprehension of PEQA-tuned LLMs, we fine-tune low-bit quantized LLMs using a instruction dataset. Our results show that even when LLMs are quantized to below 4-bit precision, their capabilities in language modeling, few-shot in-context learning, and comprehension can be resiliently restored to (or even improved over) their full-precision original performances with PEQA.

---

[*]Equal contribution

37th Conference on Neural Information Processing Systems (NeurIPS 2023).

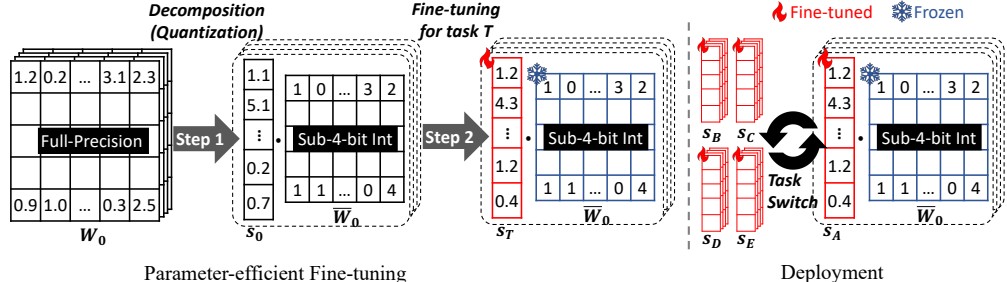

Figure 1: Illustration of our proposed PEQA scheme where $A \cdot B$ indicates the element-wise product of $A$ and $B$. PEQA is memory-efficient fine-tuning method for quantized large language models that updates only the quantization scale while keeping the integer matrix frozen. Notice a significant reduction in memory footprint when full-precision weights are converted into sub-4-bit integers.

# 1 Introduction

Large language models (LLMs) such as PaLM, LLaMA, and the GPT-series [1–7] have demonstrated unprecedented levels of task-generalization ability in various applications, including dialogue systems, question answering, summarization, and translation [8, 9]. While they can follow instructions and learn to solve tasks via in-context task descriptions or few-shot examples [10], fine-tuning allows LLMs to align their behavior with desirable traits, such as following instructions more precisely [11] or adhering to certain principles [12]. Additionally, fine-tuning can improve the scaling curve by exposing the model to large collections of task-specific instruction datasets, leading to significant performance enhancements in various unseen downstream tasks [13–17]. However, the immense computational cost of fully fine-tuning large-scale models presents challenges for researchers and developers, especially given that LLMs have billions or even trillions of parameters [18].

In response, several parameter-efficient fine-tuning (PEFT) methods have been introduced [19–21], which only update a small number of parameters compared to the pre-trained weights of LLMs. PEFT notably reduces the number of learnable parameters, making the fine-tuning of pre-trained LLMs viable by ensuring that the optimizer states' memory usage becomes negligible. These strategies lead to decreased memory usage during training and more efficient storage and seamless transitions of task-specifically fine-tuned parameters during deployment. Nonetheless, LLMs as a whole still demand significant memory, and further reductions are attainable through model compression. As outlined in Hu et al. [21], for instance, LoRA can cut the memory usage during the fine-tuning of GPT-3 175B from 1.2TB to 350GB. However, the model still requires approximately 350GB of memory for parameters in half-precision floating-point format.

Quantization is a favorable method for both compressing and accelerating neural networks by discretizing parameters into low-bit integers while maintaining a shared high-precision scale within each parameter group (e.g., channel or layer). However, during training phases, quantization-aware training (QAT) [22–25] mandates updates for all parameters, rendering it not parameter-efficient. Since post-training quantization (PTQ) [26–29] is executed after training, most existing quantization schemes primarily target the deployment phases. Although PTQ can be integrated with PEFT, when PTQ follows PEFT, the model remains intact during fine-tuning, not decreasing the memory usage. Conversely, if PTQ precedes PEFT, while there's a reduction in memory usage during fine-tuning, no inference acceleration can be achieved due to the PEFT parameters during deployment.

To bridge the gap between PEFT and quantization, we introduce the Parameter-Efficient and Quantization-aware Adaptation (PEQA), a simple yet effective quantization-aware PEFT method. As illustrated in Figure 1, PEQA encompasses two steps: (a) Decomposition (Quantization) where the parameter matrix of each fully-connected layer is decomposed into a matrix of low-bit integers and quantization scales; and (b) Fine-tuning wherein, for each downstream task, the quantization scale is fine-tuned while the integer matrix remains unchanged. For the quantized LLMs, merely updating the quantization scale leverages the advantages of PEQA. As a result, PEQA maintains the merits of PEFT, such as fewer trainable parameters, along with efficient storage and swift switching of task-specific parameters. Concurrently, it provides the benefits of quantization, including reduced

DRAM usage during both training and deployment, and inference acceleration due to fewer memory accesses at deployment.

Through this, we highlight the following:

- We introduce PEQA, a method that fine-tunes only the quantization scales of quantized LLMs, keeping the integer matrix frozen. It bridges the gap between PEFT and quantization, offering advantages such as reduced memory consumption during both training and deployment phases, seamless task transitions, and faster inference.

- To empirically validate the approach of solely fine-tuning the quantization scale while freezing the integer matrix, we compare the perplexity of LLMs fine-tuned with QAT, PEFT (+PTQ), and PEQA. The results indicate that PEQA delivers competitive performance in comparison to QAT and PEFT+PTQ, even at sub-4-bit precision.

- To assess the scalability and comprehension performance of PEQA, we apply PEQA to task-specific adaptation and instruction-tuning. Despite the reduction in model size by a factor of 4 to 5, PEQA demonstrates competitive performance up to a 65B LLM when compared to full-precision baselines. The results suggest that even when LLMs are quantized into low-bit precision, the overall comprehension capability of quantized LLMs can be effectively restored to their original performance using PEQA.

## 2   Related Work

**Large Language Models and Alignment Learning.**    Although LLMs have demonstrated great generalization capabilities through their sheer scales [2, 6, 30] and the emergent mechanism known as *in-context learning* [1], they still require significant alignment to follow natural language instructions [13], adhere to ethical guidelines or steer towards harmlessness [12], utilize external tools [31], and to be grounded in knowledge to generate truthful answers [16, 32]. In particular, instruction-tuning has been pivotal in enabling LLMs to generalize instruction-following abilities, enabling them to solve seemingly any NLP task with only the description of the task in natural text [13, 33, 34], allowing the models to be accessed in an interactive manner.

**Parameter-Efficient Fine-Tuning.**    Fine-tuning leverages the generalization capabilities elicited from the general pretraining to specialize in specific domains and tasks [35, 36] or align the LLM with target behaviors [13]. However, updating parameters in LLMs comes with a high computation cost and minimal compute environment required for gradient computation. As the hyper-scale era makes fine-tuning for LLMs prohibitively expensive, both efficient and effective alternatives to fine-tuning have received considerable attention. Specifically, inspired by the sensitivity of LLMs to prompts [37], a line of works has proposed introducing trainable prompt embeddings prepended to the input text while freezing the original LLM parameters [19, 38, 39]. As another approach, adapter modules [20] introduce task-specific parameters, which are inserted between the pre-existing layers of the model Extending on this adapter-based approach, LoRA [21] employs the concept of low-rank bottleneck modules while demonstrating comparable performance to full fine-tuning. Subsequent works have unified the various versions and diverging approaches to PEFT [40, 41] by formulating them in a single mathematical framework. These parameter-efficient methods have shown comparable performance to full model fine-tuning, presenting a cost-effective and efficient avenue for tailoring LLMs to specific tasks.

However, even with the adoption of PEFT, the inherent model size of the LLM remains a challenge to handle. One immediate solution is to apply post-training quantization (PTQ), but its interaction with task-specific parameters is still an area of active research.There have been attempts to integrate PEFT and neural network quantization, including methods like Quadapter [42] and AlphaTuning [43]. Yet, these methods have primarily been explored in smaller models of 1.3B or fewer parameters. Appendix J delineates the distinctions between our method and AlphaTuning.

**Neural Network Quantization.**    Neural network quantization consists largely of quantization-aware training (QAT) and PTQ. QAT methods [22–25] basically train not only quantization scales but all the parameters of a full-precision neural network to narrow the performance gap between the full-precision model and its quantized counterpart. Unfortunately, since QAT involves training all the weights of a full-precision network, it is not feasible to apply QAT to LLMs. To quantize LLMs, PTQ

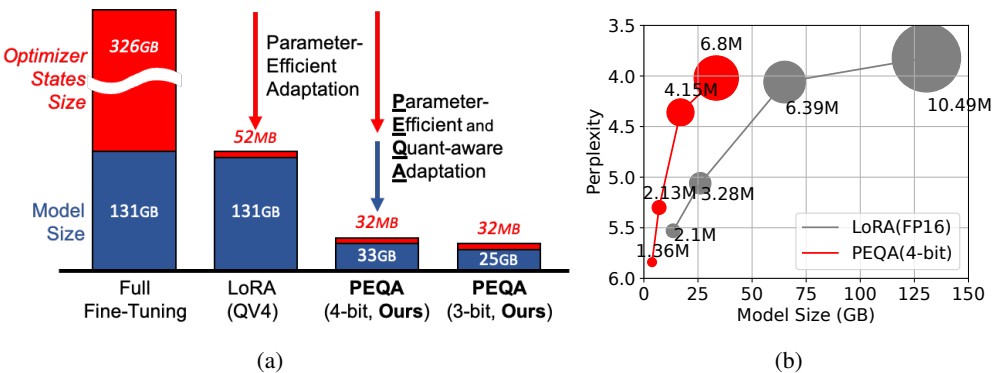

Figure 2: (a) DRAM usage comparison of LLaMA-65B on various tuning methods and (b) perplexity over model size when tuning LLaMA models with LoRA and PEQA on Wikitext2 dataset. The size of a circle indicates the number of trainable parameters. For instance, the LLaMA-65B model with LoRA has a size of 131GB and 10.49M trainable parameters. Otherwise, LLaMA-65B with 4-bit PEQA has a model size of 33GB and 6.8M trainable parameters.

techniques tailored to LLMs [26–29, 44, 45] have been presented. Although such PTQ approaches do not require learning all the parameters of an LLM at all, as PTQ occurs after training/fine-tuning LLMs, PTQ cannot allow for compressing the model size during training/fine-tuning LLMs. To reduce the model size even during training/fine-tuning LLMs, researchers have recently focused on combining PEFT with quantization.

# 3 Methodology

## 3.1 Problem Setup

**Memory Demands in Fine-tuning.** Given the significant computational demands associated with fully fine-tuning large language models, parameter-efficient fine-tuning (PEFT) methods have been introduced [19–21, 46]. One of the primary goals of PEFT methods is to reduce memory usage of the optimizer state during training, specifically by reducing the number of learnable parameters. While existing PEFT techniques do decrease memory consumption of the optimizer state and also narrow the accuracy gap between full fine-tuning and PEFT, the pre-trained weights of large language models still demand substantial memory space. When applying LoRA [21] to LLaMA-65B, for instance, even though storing optimizer states for trainable parameters consumes only 52MB (only query and value matrices are adapted with a LoRA rank of 4), the model size still occupies a huge portion of DRAM usage due to the frozen FP16 weights of a pre-trained model. To make PEFT more efficient, reducing the model size is an indispensable requisite. Given that LLMs mostly consist of fully-connected layers, compressing the weights of fully-connected layers is a key factor in compressing the model size and thus leading to more efficient PEFT.

**Inference Latency of Large Language Models.** During text generation inference, an autoregressive LLM generates tokens sequentially. A significant portion of the inference latency arises from matrix-vector multiplications, as opposed to matrix-matrix multiplications. Given that the batch size during inference is typically small [27, 28], matrix-vector multiplications tend to be memory-bound. Specifically, accessing global memory, such as DRAM, is expensive on contemporary high-end GPUs. Thus, the number of weights loaded into registers profoundly impacts the speed of multiplication between a matrix and a vector. To decrease the number of weights (subsequently increasing the weights loaded into registers), quantization is a widely researched method for both compressing and speeding up neural networks [29, 44, 47]. While quantization-aware training (QAT) imposes a significant load on both computation and memory, post-training quantization (PTQ) is often viewed as a fallback strategy among traditional quantization techniques to enhance the generation latency of LLMs. To both accelerate LLM inference and retain all advantages of PEFT, an innovative alternative approach should be pursued.

Table 1: Comparison of PEQA with other methods using LLaMA 65B on the DRAM usage and training time during fine-tuning, the DRAM storage for deployment, the inference acceleration, and task-switching efficiency. The DRAM usage estimation for PEFT is based on LoRA. PEFT+PTQ denotes PTQ after PEFT and PTQ+PEFT denotes PTQ before PEFT.

| Method | DRAM (Fine-Tuning) | DRAM (Deployment) | Inference Speed | Task-Switching |
|---|---|---|---|---|
| Full Fine-Tuning | 457GB | 131GB | Slow | Slow |
| PEFT | 131GB | 131GB | Slow | Fast |
| PEFT+PTQ | 131GB | 33GB | Fast | Slow |
| PTQ+PEFT | 33GB | 33GB | Slow | Fast |
| **PEQA (Ours)** | **33GB** | **33GB** | **Fast** | **Fast** |

## 3.2 Parameter-Efficient and Quantization-aware Adaptation (PEQA)

Quantization reduces bit-precision for inference acceleration, less storage and increasing throughput. INT8 quantization, which lower the bit-precision for both activations and weights, utilize dedicated engine to effectively accelerate arithmetic computation [48]. This is effective for large batches where computing speed matters but less so for smaller batches constrained by memory. To tackle this memory issue, weight-only quantization keeps high precision for activations (e.g., FP16) but compresses weights to 4-bit or less, targeting memory I/O enhancement in modern GPUs [28, 47]. For simplicity, we mainly focus on low-bit weight-only quantization in a linear asymmetric per-channel context in this paper.

For pre-trained weights of a fully-connected layer $\boldsymbol{W}_0 \in \mathbb{R}^{n \times m}$, while PEQA can be applied to quantized LLMs, we first quantize $\boldsymbol{W}_0$. In other words, for a given bit-width $b$, quantized pre-trained weights $\widehat{\boldsymbol{W}}_0$ can be written as

$$\widehat{\boldsymbol{W}}_0 = \boldsymbol{s}_0 \cdot \overline{\boldsymbol{W}}_0 = \boldsymbol{s}_0 \cdot \left( \text{clamp}\left( \left\lfloor \frac{\boldsymbol{W}_0}{\boldsymbol{s}_0} \right\rceil + \boldsymbol{z}_0, 0, 2^b - 1 \right) - \boldsymbol{z}_0 \right), \tag{1}$$

where $A \cdot B$, $\lfloor \cdot \rceil$, and $\text{clamp}(\cdot, a, b)$ indicate the element-wise product of $A$ and $B$, the rounding function, and the clamping function into the range $[a, b]$, respectively, while per-channel scales and zero-points (namely, $\boldsymbol{s}_0, \boldsymbol{z}_0 \in \mathbb{R}^{n \times 1}$) are initialized to minimize $\|\boldsymbol{W}_0 - \widehat{\boldsymbol{W}}_0\|_F^2$. Notice that $\boldsymbol{s}_0$ and $\boldsymbol{z}_0$ are not related to any downstream task. Here, we freeze $\overline{\boldsymbol{W}}_0 = \text{clamp}\left( \left\lfloor \frac{\boldsymbol{W}_0}{\boldsymbol{s}_0} \right\rceil + \boldsymbol{z}_0, 0, 2^b - 1 \right) - \boldsymbol{z}_0$, which is the integer quantization indices of $\boldsymbol{W}_0$, for every full-connected layer in a pre-trained LLM. And then we fine-tune only $\boldsymbol{s}_0$ (residing outside the clamp function in Eq. 1) while sharing $\overline{\boldsymbol{W}}_0$ across all downstream tasks. Consequently, quantized pre-trained weights $\widehat{\boldsymbol{W}}_0$ are adapted to a downstream task as follows:

$$\widehat{\boldsymbol{W}} = (\boldsymbol{s}_0 + \Delta\boldsymbol{s}) \cdot \overline{\boldsymbol{W}}_0 = (\boldsymbol{s}_0 + \Delta\boldsymbol{s}) \cdot \left( \text{clamp}\left( \left\lfloor \frac{\boldsymbol{W}_0}{\boldsymbol{s}_0} \right\rceil + \boldsymbol{z}_0, 0, 2^b - 1 \right) - \boldsymbol{z}_0 \right), \tag{2}$$

where $\Delta\boldsymbol{s} \in \mathbb{R}^{n \times 1}$ represents the gradient update of $\boldsymbol{s}_0$ obtained by adaptation to a downstream task. We dub Eq. 2 as Parameter-Efficient and Quantization-aware Adaptation (PEQA). PEQA is a memory-efficient fine-tuning method dedicated to quantized LLMs by solely updating quantization scales $\boldsymbol{s}_0$. With $\overline{\boldsymbol{W}}_0$ being frozen and shared for all downstream tasks, $\boldsymbol{s}_0 + \Delta\boldsymbol{s}$ are task-specific parameters in PEQA, which can be quickly and easily swapped when it is needed to switch to a different downstream task. Note that, PEQA can be seamlessly applied not only to weight-only quantized LLMs but also to weight-activation quantized ones. The overall procedure of PEQA is described in Figure 1 in detail.

## 3.3 Benefits of PEQA Inherited from Bridging the Gap between PEFT and Quantization

PEQA is designed to have the advantages of both existing PEFT methods [19, 21, 46] and quantized LLM [28, 44, 47, 49]. We summarize the benefits of PEQA in this subsection.

Table 2: To empirically confirm the validity of PEQA's approach, we compare the perplexity (PPL) of fine-tuned LLMs through QAT, PEFT+PTQ, and PEQA on Wikitext2 [51] for GPT-Neo 2.7B, GPT-J 6B, LLaMA 7B, and LLaMA 13B. Weights are quantized into either 3-bit or 4-bit per channel, without a group size [28, 49]. LoRA configuration is set to QV4. The lower PPL, the better.

| Method | W Bits | GPT-Neo 2.7B | GPT-J 6B | LLaMA 7B | LLaMA 13B |
|---|---|---|---|---|---|
| QAT | 4 | 11.07 | 8.81 | 5.76 | 5.26 |
| LoRA + OPTQ | 4 | 12.09 | 8.91 | 7.13 | 5.31 |
| PEQA (Ours) | 4 | 11.38 | 8.84 | 5.84 | 5.30 |
| QAT | 3 | 12.37 | 9.60 | 6.14 | 5.59 |
| LoRA + OPTQ | 3 | 21.93 | 11.22 | 19.47 | 7.33 |
| PEQA (Ours) | 3 | 12.54 | 9.36 | 6.19 | 5.54 |

**Benefits of PEFT.** By solely updating the quantization scales, PEQA substantially reduces the memory overhead associated with optimizer state, a feature consistent with other PEFT approaches. Notably, the utilization of quantization scales $s_0 + \Delta s$ allows PEQA to swiftly and effortlessly switch between task-specific parameters. This capability positions PEQA as ideally suited for deployment of quantized LLMs as a service, mirroring another key advantage of earlier PEFT methods. Note, however, that such a capability is not present in PEFT+PTQ (i.e., the case where PTQ is applied after PEFT) due to the non-reversible quantizers, such as the rounding function.

**Benefits of Quantization.** Previous PEFT methods freeze the pre-trained weights $W_0$ and utilize additional learnable parameters to reduce the memory usage of optimizer state. Similarly, PEQA freezes quantized pre-trained weights $\overline{W}_0$ (which are the integer quantization values of $W_0$) and fine-tunes quantization scales $s_0$. Since $\overline{W}_0$ is a $b$-bit integer matrix, not only can PEQA reduce the optimizer states' size but also the model size, leading to even greater efficiency in the PEFT scheme, as illustrated in Figure 2a. In addition, since $\widehat{W}$ is a $b$-bit quantized matrix, PEQA can speed up token generation process at inference through dedicated kernels that accelerate the multiplication between a quantized weight matrix and a half-precision activation vector, as described by Frantar et al. [28], Lin et al. [47], and Park et al. [49]. It is worth noticing that employing PTQ before fine-tuning (PTQ+PEFT) [50] allows for memory-efficient fine-tuning and seamless task transition for the quantized LLM; however, PTQ+PEFT is not able to inherit from inference acceleration of quantization. As a result, PEQA can achieve both model compression in the process of fine-tuning and inference acceleration for fine-tuned models with marginal performance degradation compared to LoRA, one of the state-of-the-art PEFT techniques, as shown in Figure 2b.

The comparison of PEQA with other methods using the LLaMA 65B is summarized in Table 1.

## 4 Experiments

In this section, we empirically validate the effectiveness of our proposed PEQA method by examining its performance in both parameter-efficient fine-tuning (PEFT) and as a quantization method. We achieve this goal by using a series of benchmarks [52–57], datasets [51, 58, 59], and LLMs [4, 6, 60, 61] that have been publicly introduced. In Section 4.1, to empirically confirm the validity of fine-tuning only the quantization scale while freezing the integer matrix, we compare the perplexity of fine-tuned LLMs through quantization-aware training (QAT), PEFT (+PTQ), and PEQA. In Section 4.2, to evaluate PEQA's scalability and task-specific adaptation performance, we fine-tune and assess LLMs on the Wikitext2 [51] and PennTreeBank [58] datasets using PEQA and LoRA [21]. Section 4.3 is dedicated to showcasing PEQA's performance-restoring capability through instruction-tuning on the Alpaca [59] dataset after round-to-nearest (RTN) quantization over the full-precision original model.

To assess PEQA's performance as a PEFT method, we contrast PEQA with LoRA, which is currently recognized as one of the leading PEFT methods. As discussed in 3.1, we employ a baseline case that merges OPTQ [28], the state-of-the-art weight-only post-training quantization (PTQ) method for LLMs, with LoRA in order to evaluate PEQA's quantization capabilities. In the context of LoRA, QV4 signifies the application of query and value layer weights with a LoRA rank of 4, while QKVO16

Table 3: To show scalability of PEQA, the perplexity (PPL) on Wikitext2 and PennTreeBank (PTB) was compared with LoRA and PEQA. In this comparison, only the weights were quantized into 3-bit and 4-bit per-channel without group size. LoRA configuration is set to QV4. A lower PPL value indicates better performance.

| Method | W Bits | GPT-Neo 2.7B | GPT-J 6B | LLaMA 7B | LLaMA 13B | LLaMA 30B | LLaMA 65B |
|---|---|---|---|---|---|---|---|
| | | | | Wikitext2 | | | |
| LoRA | 16 | 10.63 | 8.50 | 5.53 | 5.06 | 4.06 | 3.82 |
| LoRA+OPTQ | 4 | 12.09 | 8.91 | 7.13 | 5.31 | 4.39 | 4.10 |
| PEQA (Ours) | 4 | **11.38** | **8.84** | **5.84** | **5.30** | **4.36** | **4.02** |
| LoRA+OPTQ | 3 | 21.93 | 11.22 | 19.47 | 7.33 | 5.94 | 5.32 |
| PEQA (Ours) | 3 | **12.54** | **9.36** | **6.19** | **5.54** | **4.58** | **4.27** |
| | | | | PTB | | | |
| LoRA | 16 | 15.92 | 12.92 | 9.14 | 8.52 | 7.21 | 7.11 |
| LoRA+OPTQ | 4 | 18.83 | 13.46 | 11.22 | 8.83 | **7.55** | 7.46 |
| PEQA (Ours) | 4 | **16.55** | **13.30** | **9.69** | **8.64** | 7.68 | **7.36** |

indicates the application of query, key, value, and output projection layer weights with a LoRA rank of 16. For PEQA, we utilize round-to-nearest (RTN) for the initialization method of quantized LLM.

## 4.1 Comparing Quantization Capabilities: PEQA vs. QAT vs. PEFT+PTQ

Table 2 presents the perplexity when various quantized LLMs are fine-tuned using QAT, PEFT (+PTQ), and PEQA. To validate our approach, PEQA, which solely fine-tunes the quantization scale as outlined in Eq. 2 while simultaneously maintaining the integer matrix in a frozen state, we use QAT as an upper bound and PEFT+PTQ as a lower bound. Note that QAT, unlike PEQA, updates all parameters including pre-trained weights as well as quantization scales. Table 2 reveals the competitive performance of PEQA compared to QAT. Furthermore, our observations indicate that PEQA consistently outperforms the combination of LoRA and OPTQ for any selected model, regardless of whether a 3-bit or 4-bit setting is employed. Such superior performance can be attributed to PEQA's method of fine-tuning quantized LLMs, which minimizes the final task loss on the full training data, a capability that OPTQ lacks. Detailed settings are in Appendix B.

Diving deeper into the comparison between QAT and PEQA, it is important to note that QAT minimizes the final task loss computed from a weight-only quantized model, as described in Eq. 1, with respect to both $W_0$ and $s_0$. Note that QAT includes all pre-trained weights for training, resulting in the practical model size limitation of LLMs under investigation being capped at 13B in our experiments. Despite the fact that QAT also updates $W_0$ in Eq. 1, which is one of the most simple and straightforward approach though, we observe that the performance gap between QAT and PEQA narrows when the 4-bit association is introduced, especially as the size of LLMs increases. Impressively, PEQA can even outperform QAT in a 3-bit setting, a notably low-bit setting that challenges OPTQ in terms of quantizing LLMs. These findings suggest that the approach of PEQA, solely updating quantization scales while freezing the integer quantization values of pre-trained weights, can achieve performance comparable to that of QAT.

## 4.2 Task-specific Adaptation with Wikitext2 and PennTreeBank Datasets

**Task-specific Adaptation and Scalability.** We evaluate the task-specific adaptation performance and scalability of PEQA by employing GPT-Neo, GPT-J, and LLaMA models (up to 65B) on the Wikitext2 [51] and PennTreeBank (PTB) [58] datasets. The adaptation performance of PEQA is compared with LoRA, with its configuration set to QV4. As depicted in Table 3, we note a gradual convergence of PEQA's perplexity to that of full-precision LoRA, with only marginal PPL degradation as the model size expands. Thus, Table 3 demonstrates that PEQA, compared to a prominent PEFT technique that utilizes full-precision pre-trained language model (PLM), can maintain a competitive perplexity level in LLMs while concurrently reducing DRAM usage through low-bit quantized

Table 4: Number of learnable parameters and model size of GPT-Neo, GPT-J and LLaMAs. PEQA configuration is set to 4-bit or 3-bit channel-wise quantization.

|  | Method | GPT-Neo 2.7B | GPT-J 6B | LLaMA 7B | LLaMA 13B | LLaMA 30B | LLaMA 65B |
|---|---|---|---|---|---|---|---|
| # of | LoRA (QV4) | 1.31 | 1.84 | 2.10 | 3.28 | 6.39 | 10.49 |
| Learnable | LoRA (QKVO16) | 5.24 | 7.34 | 8.39 | 13.11 | 25.56 | 41.94 |
| Param. (M) | PEQA (Ours) | 0.74 | 1.03 | 1.36 | 2.13 | 4.15 | 6.80 |
| Model | LoRA (QV4) | 5.30 | 12.10 | 13.48 | 26.03 | 65.06 | 130.57 |
| Size | PEQA (Ours, 4-bit) | 1.53 | 3.65 | 3.77 | 7.01 | 16.92 | 33.45 |
| (GB) | PEQA (Ours, 3-bit) | 1.21 | 2.94 | 2.96 | 5.42 | 12.90 | 25.35 |

Table 5: Multi-scale (grouping) performance with PEQA-tuned LLaMA 7B and 13B on Wikitext2 where $g$ indicates the group size [49]. The perplexity consistently increases as PEQA take on more learnable parameters.

| Model | W Bits | Channel-Wise | $g256$ | $g128$ | $g64$ |
|---|---|---|---|---|---|
| LLaMA 7B | 4 | 5.84 | 5.69 | 5.66 | 5.64 |
|  | 3 | 6.19 | 5.96 | 5.91 | 5.89 |
| LLaMA 13B | 4 | 5.30 | 5.18 | 5.16 | 5.16 |
|  | 3 | 5.54 | 5.40 | 5.37 | 5.34 |

weights. Notably, for a 3-bit quantization, PEQA experiences less performance degradation as the model size decreases due to extreme low-bit quantization compared to the combined LoRA and OPTQ. To further elucidate our findings, we have provided figures illustrating the results of 3-bit and 4-bit PEQA in the Appendix D. The comprehensive results indicate that for the deployment stage, PEQA allows models with larger parameters to operate under DRAM usage constraints, outperforming full-precision PEFT methods. For instance, under a restricted DRAM footprint, large LLaMA models can be explored using PEQA, while full-precision LoRA permits only smaller LLaMA models. Additional results with OPT [4], ranging from 1.3B to 66B models are included in the Appendix E. The detailed experimental settings are also included in the Appendix C.

**Model Size and Number of Learnable Parameters.**    In an effort to estimate the DRAM usage necessitated by PEQA and LoRA during training and deployment, we outline the number of learnable parameters (for training) and the model size (expressed in gigabytes, GB, for deployment) in Table 4. As demonstrated in Table 4, PEQA involves fewer learnable parameters than LoRA when a quantization scale is assigned to each channel of pre-trained weights. For instance, PEQA has approximately 1.54 times fewer learnable parameters for LLaMA models than LoRA (QV4). In addition to having fewer learnable parameters, PEQA, through low-bit weight quantization, can also reduce the model size, which captures a huge amount of the DRAM footprint in fine-tuning LLMs. Remarkably, when fine-tuning LLaMA 30B using PEQA with 4-bit precision, the resulting model size is significantly smaller than that obtained by adapting 13B through LoRA, and slightly larger than the model adapted from LLaMA 7B using LoRA. Additionally, a comparison of the memory peak during training between PEQA and LoRA is provided in Appendix L.

**Group-wise Quantization.**    Group-wise per-channel quantization [49], where weight groups in channels share quantization parameters, maintains accuracy at lower bits. In Table 5, we present that the performance incrementally improves as more learnable parameters are incorporated into PEQA. In particular, for Table 5, we examine various group sizes (denoted by $g$) when quantizing the weights [49, 62]. Through relatively straightforward grouping (employed to regulate the number of learnable parameters for PEQA), the perplexity incrementally decreases as more learnable parameters are utilized. Detailed settings are in Appendix G.

Table 6: Common-sense reasoning and in-context learning performance of parameter-efficient instruction-tuned LLaMAs [6] using Alpaca datasets. LoRA configuration is set to QKVO16. Quantization precision of PEQA is set to 4-bit per-channel without group size. Note that ARC-C, ARC-E and OBQA stands for ARC-Challenge, ARC-Easy, and OpenBookQA respectively.

| Method | # Params | Model Size (GB) | PIQA | HellaSwag | ARC-C | ARC-E | OBQA | Average |
|---|---|---|---|---|---|---|---|---|
| | | | | Zero-Shot | | | | |
| LLaMA | 7B | 13.5GB | 77.3 | 73.0 | 41.4 | 52.5 | 42.4 | 57.3 |
| | 13B | 26.1GB | 79.1 | 76.2 | 44.5 | 59.9 | 42.2 | 60.4 |
| | 30B | 65.1GB | 80.1 | 79.2 | 45.5 | 58.9 | 42.0 | 61.1 |
| + LoRA | 7B | 13.5GB | 78.6 | 73.3 | 43.7 | 55.8 | 43.0 | 58.9(+1.6) |
| | 13B | 26.1GB | 79.6 | 76.7 | 46.3 | 62.0 | 43.2 | 61.5(+1.1) |
| | 30B | 65.1GB | 81.8 | 80.3 | 48.2 | 61.6 | 42.8 | 62.9(+1.8) |
| + PEQA | 7B | **3.8GB** | 77.9 | 71.4 | 42.4 | 57.2 | 42.0 | 58.2(+0.9) |
| | 13B | **7.0GB** | 78.9 | 74.0 | 46.4 | 62.5 | 42.8 | 60.9(+0.5) |
| | 30B | **16.9GB** | 80.3 | 78.4 | 49.8 | 63.3 | 42.8 | 62.9(+1.8) |
| | | | | Five-Shot | | | | |
| LLaMA | 7B | 13.5GB | 79.4 | 75.3 | 45.6 | 65.8 | 44.0 | 62.0 |
| | 13B | 26.1GB | 80.0 | 78.4 | 50.4 | 70.8 | 47.2 | 65.4 |
| | 30B | 65.1GB | 82.5 | 82.2 | 56.2 | 74.9 | 47.0 | 68.6 |
| + LoRA | 7B | 13.5GB | 79.9 | 75.2 | 46.4 | 66.5 | 47.2 | 63.0(+1.0) |
| | 13B | 26.1GB | 81.1 | 78.8 | 53.5 | 72.4 | 47.0 | 66.6(+1.1) |
| | 30B | 65.1GB | 84.1 | 83.3 | 59.5 | 79.2 | 50.6 | 71.4(+2.8) |
| + PEQA | 7B | **3.8GB** | 78.9 | 73.2 | 45.1 | 65.4 | 44.0 | 61.3(−0.7) |
| | 13B | **7.0GB** | 80.7 | 76.0 | 50.9 | 71.6 | 48.0 | 65.5(+0.1) |
| | 30B | **16.9GB** | 82.7 | 80.2 | 56.8 | 75.5 | 47.6 | 68.6(+0.0) |

## 4.3 Instruction-tuning with the Alpaca Dataset

Although inference cost and training efficiency are crucial, the methods of PEFT and quantization have not been extensively explored. While LoRA and PEQA have been evaluated on adaptation performance for task-specific purposes as discussed in Section 4.2, it has not been widely corroborated that fine-tuning via PEFT can retain the performance for unseen tasks. Thus, given that RTN quantization results in a non-negligible performance degradation in LLMs, it is important to assess how much the performance of low-bit quantized LLMs is degraded on comprehensive tasks. To address these concerns, we conduct comprehensive experiments, benchmarking our techniques on prevalent instruction-following datasets and assessing the response quality of PEQA-tuned LLMs. Furthermore, to determine if PEQA can regain the performance of full-precision LLMs, we employ RTN quantization in conjunction with PEQA instruction-tuning across LLaMAs.

**Experimental Settings.** We train LLaMAs [6, 7] in various sizes on the Alpaca dataset [59], which is one of the popular instruction-following datasets generated from outputs of InstructGPT [11]. Then, we test the models on other downstream tasks such as common-sense reasoning tasks [52–55] and massive multitask language understanding (MMLU) [56]. Due to limited time and resources, we could not conduct an exhaustive search over hyper-parameters such as the learning rate or epoch. Instead, we followed the training recipe from Taori et al. [59]. The LoRA configuration is set to QKVO16. Detailed settings can be found in Appendix H.

**Common-Sense Reasoning.** We conducted an experiment on five tasks [52–55] to assess whether the performance of common-sense reasoning and in-context learning can be sustained even after instruction-tuning LLMs on the Alpaca dataset via LoRA or PEQA. As depicted in Table 6, the results show that LLMs fine-tuned with LoRA or PEQA maintain a consistent trend in common-sense reasoning tasks. Furthermore, since PEQA's performance aligns closely with that of full-precision

Table 7: Massive Multitask Language Understanding (MMLU) benchmark performance of PEQA-tuned LLaMAs using Alpaca datasets. Five-shot accuracy is reported for the MMLU. Quantization precision of PEQA is set to 4-bit. When we quantize LLaMA [6] into 4-bit precision using the RTN method, no group size is applied. For LLaMA2 [7], a group size of 256 is used with the RTN method. Note that RTN stands for round-to-nearest in the table.

|  | # Params | Model Size | Humanities | STEM | Social Sciences | Other | Average |
|---|---|---|---|---|---|---|---|
| LLaMA [6] | 7B | 13.5GB | 32.6 | 29.6 | 38.0 | 37.9 | 34.4 |
|  | 13B | 26.1GB | 42.8 | 36.1 | 53.3 | 53.2 | 46.1 |
|  | 30B | 65.1GB | 54.6 | 46.5 | 66.1 | 63.4 | 57.4 |
| + RTN | 7B | **3.8GB** | 28.4 | 25.6 | 26.9 | 31.8 | 28.3 |
| (w/o group size) | 13B | **7.0GB** | 30.5 | 27.2 | 35.5 | 38.8 | 32.8 |
|  | 30B | **16.9GB** | 39.6 | 34.0 | 46.1 | 49.7 | 42.1 |
| + PEQA | 7B | **3.8GB** | 35.7 | 30.9 | 38.2 | 40.0 | 35.8 |
|  | 13B | **7.0GB** | 42.8 | 37.7 | 53.6 | 49.0 | 45.0 |
|  | 30B | **16.9GB** | 51.1 | 44.1 | 62.4 | 60.7 | 54.3 |
| LLaMA2 [7] | 7B | 13.5GB | 43.3 | 37.0 | 51.8 | 52.4 | 45.9 |
|  | 13B | 26.0GB | 54.4 | 44.2 | 63.4 | 60.8 | 55.7 |
|  | 70B | 138.0GB | 65.2 | 57.9 | 80.3 | 74.7 | 69.1 |
| + RTN | 7B | **3.8GB** | 39.5 | 35.5 | 49.3 | 49.9 | 43.2 |
| (g256) | 13B | **7.0GB** | 50.2 | 42.6 | 61.3 | 59.7 | 53.2 |
|  | 70B | **35.3GB** | 63.7 | 55.9 | 78.4 | 71.6 | 67.0 |
| + PEQA | 7B | **3.8GB** | 52.0 | 38.4 | 54.1 | 52.0 | 48.1 |
|  | 13B | **7.0GB** | 60.5 | 45.0 | 63.3 | 57.0 | 55.3 |
|  | 70B | **35.3GB** | 73.9 | 55.3 | 77.8 | 68.2 | 67.5 |

adaptation, this consistency is observed even when the model size has been reduced through low-bit weight quantization. We utilized the evaluation code from Eleuther AI's *lm-evaluation-harness* [63].

**Massive Multitask Language Understanding.** To assess whether the performance of PEQA-tuned models can be restored to the levels of full-precision model's performance, starting from RTN performance, we test our models on the MMLU benchmark containing 57 multiple-choice problems across various domains and levels of knowledge [56]. In this experiment, we utilize the RTN results as a baseline to determine the extent of degradation on quantized LLM. As shown in Table 7, instruction-tuning with PEQA boosts the performance of RTN quantized models. This observation supports our claim that our approach enables LLMs to regain their few-shot in-context learning and understanding capabilities, even though they are significantly smaller than their original model size through quantization. Unfortunately, it seems that the PEQA-tuning does not achieve the best performance in fine-tuning larger models. This might be because PEQA-tuning did not been sufficiently explored different epochs or learning rates. Nonetheless, the observation that the performance of the quantized LLaMAs is restored through PEQA-tuning using an instruction-following dataset highlights the potential to further enhance the accuracy of PTQ methods.

## 5 Conclusion

Fine-tuning aligns large language models (LLMs) with specific purposes. To maintain the comprehensive capabilities of LLMs while effectively aligning them, we introduce PEQA, a method that seamlessly combines the advantages of parameter-efficient fine-tuning (PEFT) and quantization in LLMs. PEQA not only reduces DRAM consumption during fine-tuning but also accelerates inference latency for deployment by retaining weights in a low-bit quantized format. Through rigorous testing across various datasets and LLMs, we have found that PEQA can match the performance of full-precision baselines in task-specific adaptations, even with a significant reduction in model size. When combined with instruction-tuning, PEQA's performance demonstrates its ability to both preserve and enhance comprehensive knowledge after the inherent compromises of quantization, recovering the performance of original model by simply updating the quantization scales of the quantized LLM.

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

## A  Common Experimental Settings

For the common experimental settings, AdamW [64] optimizer and linear-decaying learning rate scheduler were used. We use Deepspeed repository [65] [2] for FP16 and BF16 training. Additionally, we utilize Huggingface repository[66][3] for training, evaluation code and dataset.

## B  Experimental Settings of Section 4.1

We compare the perplexity when weights are quantized and adapted by quantization-aware training (QAT), LoRA with post-training quantization (PTQ), and PEQA, using the Wikitext2 dataset in Section 4.1. The LoRA configuration is set to QV4. For PTQ method, we utilize OPTQ [28] [4] which is state-of-the-art low-bit weight-only PTQ method. We set the model's maximum sequence length to 1024. Batch size and epoch for all experiments are set to 128 and 15 respectively. The learning rates for the experiments of Table 2 are displayed in Table 8. Learning rates for LoRA and PEQA are shown in Appendix C.

Table 8: Learning rates of QAT in Table 2.

| Method | W Bits | GPT-Neo 2.7B | GPT-J 6B | LLaMA 7B | LLaMA 13B |
|--------|--------|--------------|----------|----------|-----------|
| QAT    | 4      | 4e-5         | 5e-6     | 1e-5     | 3e-5      |
| QAT    | 3      | 6e-5         | 1e-5     | 2e-5     | 1e-5      |

## C  Experimental settings of Table 3

In Section 4.2, Table 3 show the scalability and task-specific adaptation performance of PEQA by comparing with LoRA and LoRA+OPTQ on Wikitext2 [51] and PennTreeBank (PTB) [58] datasets. Detailed experimental settings are as follows. LoRA configuration is set to QV4. For PTQ method, we utilize OPTQ [28] which is state-of-the-art low-bit weight-only PTQ method. We set input sequence length after tokenization (block size) to 1024 for under 65B models. For LLaMA 65B, input sequence length after tokenization is set to 768 due to memory issue. Batch size and epoch for all experiments are set to 128 and 15 respectively. Learning rates for Table 3 experiments are shown in Table 9.

Table 9: Learning rate of LoRA and PEQA in Table 3 on Wikitext2 and PTB datasets.

| Method | W Bits | GPT-Neo 2.7B | GPT-J 6B | LLaMA 7B | LLaMA 13B | LLaMA 30B | LLaMA 65B |
|--------|--------|--------------|----------|----------|-----------|-----------|-----------|
| | | | Wikitext2 | | | | |
| LoRA | 16 | 5e-4 | 6e-4 | 1e-4 | 1e-4 | 2e-4 | 4e-5 |
| PEQA (Ours) | 4 | 5e-5 | 6e-6 | 6e-6 | 1e-5 | 1e-5 | 1e-5 |
| PEQA (Ours) | 3 | 6e-5 | 5e-5 | 2e-5 | 6e-5 | 3e-5 | 3e-5 |
| | | | PTB | | | | |
| LoRA | 16 | 2e-3 | 1e-3 | 8e-4 | 5e-4 | 4e-4 | 6e-4 |
| PEQA (Ours) | 4 | 3e-4 | 5e-5 | 5e-5 | 5e-5 | 3e-5 | 6e-5 |

---

[2] https://github.com/microsoft/DeepSpeed
[3] https://github.com/huggingface/transformers/tree/main/examples/pytorch/language-modeling
[4] https://github.com/IST-DASLab/gptq

# D   The Perplexity of $3$-bit and $4$-bit PEQA on Wikitext2 Dataset

Figure 3 illustrates the results of 3-bit and 4-bit PEQA's next token prediction performance on the Wikitext2 dataset. As shown in Figure 3, 3-bit performance of PEQA shows lower perplexity than 3-bit post-training quantized LoRA. The results from the 3-bit PEQA show that PEQA allows for continuity in model size options under DRAM usage constraints.

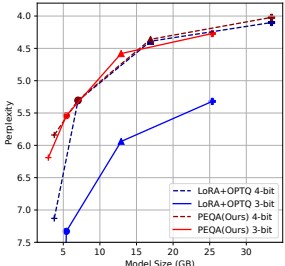

Figure 3: The perplexity over model size of 3/4-bit performance of PEQA and LoRA+OPTQ

# E   OPT Models Adapted with PEQA and LoRA on Wikitext2 Dataset

Table 10 shows the perplexity of OPT[4] models adapted with PEQA and LoRA on the Wikitext2 dataset. The perplexity gap between LoRA and PEQA becomes smaller as the model size increases.

Table 10: The perplexity (PPL) on Wikitext2 for OPT 1.3B to 66B. In this comparison, only the weights were quantized into $4$-bit. A lower PPL value indicates better performance.

| Method | W Bits | OPT $1.3$B | OPT $2.7$B | OPT $6.7$B | OPT $13$B | OPT $30$B | OPT $66$B |
|--------|--------|-----------|-----------|-----------|----------|----------|----------|
| LoRA(QV4) | 16 | 11.58 | 10.25 | 8.96 | 8.44 | 7.93 | 7.64 |
| PEQA(Ours) | 4 | 12.40 | 10.78 | 9.34 | 8.74 | 8.11 | 7.86 |

# F   LoRA Configuration Comparison on Wikitext2 Dataset

As shown in Table 11, the LoRA target module configuration of QV$4$ and QKVO$16$ has not much effect on perplexity on Wikitext2 experimental results. Table 11 shows equal tendency as mentioned in [21]. We utilize QV$4$ configuration for Section 4.2 and QKVO$16$ configuration for Section 4.3 respectively.

Table 11: The perplexity (PPL) on Wikitext2 was compared with LoRA QV$4$ and QKVO$16$. A lower PPL value indicates better performance.

| Method | # Bits | GPT-Neo 2.7B | GPT-J 6B | LLaMA 7B | LLaMA 13B | LLaMA 30B | LLaMA 65B |
|--------|--------|-------------|----------|----------|-----------|-----------|-----------|
| LoRA(QV4) | 16 | 10.63 | 8.50 | 5.53 | 5.06 | 4.06 | 3.82 |
| LoRA(QKVO16) | 16 | 10.67 | 8.50 | 5.50 | 5.06 | 4.06 | 3.81 |

# G Experimental Settings of Multi-scale Performance

In Section 4.2, Table 5 shows the perplexity of PEQA with grouping learnable parameters. We set model maximum sequence length to 1024. Batch size and epoch for all experiments are set to 128 and 15 respectively. Learning rates for experiments are shown in Table 12.

Table 12: Learning rate for Table 5.

| Model | W Bits | $g-1$ | g256 | g128 | g64 |
|---|---|---|---|---|---|
| LLaMA 13B | 4 | 1e-5 | 4e-5 | 4e-5 | 3e-5 |
|  | 3 | 6e-5 | 9e-5 | 9e-5 | 5e-5 |
| LLaMA 7B | 4 | 6e-6 | 2e-5 | 2e-5 | 1e-5 |
|  | 3 | 2e-5 | 6e-5 | 4e-5 | 7e-5 |

# H Experimental Settings of Section 4.3

In Section 4.3, we use the Alpaca dataset [59] for instruction-tuning. We set learning rate, epoch, and quantization group size as in Table 13. The batch size is set to 128 for all experiments in this subsection. As mentioned in Section 4.3, due to limited time and resources, we couldn't conduct an exhaustive search over hyper-parameters such as learning rate or epoch. We believe that there are hyper-parameters that can perform better. For LLaMA 1 series (LLaMA 7, 13, and 30B), we truncate the prompt to the length of 2024 since their maximum sequence length is 2024 when evaluating the massive multitask lanugage understanding (MMLU) benchmark. Thus, for LLaMA2-70B, we set the tokenizer max length to 1024 on fine-tuning due to the resource limit. Otherwise, we use default max length of tokenizer[5] on training. For the evaluation, we use default tokenizer setting. For every experiment in this section, the configuration of PEQA is set to 4-bit RTN quantization.

Table 13: The learning rate, epoch, quantization group size [49] for experiments on Section 4.3. the weights were quantized into 4-bit.

| Hyper-parameter | LLaMA 7B | LLaMA 13B | LLaMA 30B | LLaMA2 7B | LLaMA2 13B | LLaMA2 70B |
|---|---|---|---|---|---|---|
| Epoch | 3 | 3 | 5 | 3 | 3 | 5 |
| Learning rate | 2e-5 | 2e-5 | 5e-6 | 5e-6 | 5e-6 | 5e-6 |
| Group size | Per-channel | Per-channel | Per-channel | 256 | 256 | 256 |

---

[5]e.g. 'hf-internal-testing/llama-tokenizer' for LLaMA-1 series, 'meta-llama/Llama-2-70b-hf' for LLaMA-2 series.

# I LLaMA 7B and 13B on Natural Instruction

To evaluate the instruction-following ability of instruct-tuned models, we test them on another instruction-following dataset, Natural Instruction (NI) [57]. Different from the Alpaca dataset, instructions of NI were collected by humans for existing 61 NLP tasks. For simplicity, we utilize evaluation splits consisting of 12 subtasks and restrict the maximum number of instances for each task to 200. At test time, the model should generate proper output for the given input with instruction for the target unseen task. As shown in Table 14, we find LLaMAs trained with PEQA show consistently better zero-shot task generalization performance (ROUGE-L) in NI for all parameter sizes compared to those from LoRA.

Table 14: Natural Instruction benchmark performance of parameter-efficient instruction-tuned LLaMAs using Alpaca datasets. Zero-shot performance (ROUGE-L) is reported for the NI. LoRA configuration is set to QKVO16. Quantization precisions of LoRA w/ OPTQ and PEQA are set to 4-bit.

| # Params | LLaMA | +LoRA | +LoRA w/OPTQ | +PEQA |
|---|---|---|---|---|
| 7B | 9.4 | 24.4 | 25.0 | **27.1** |
| 13B | 8.9 | 31.3 | 29.2 | **34.1** |

# J Comparison with AlphaTuning

When diving deeper into quantization scales, learnable parameters for both PEQA and AlphaTuning, it's worth noting that PEQA's adherence to uniform quantization means there's only one shared quantization scale for integer weight. Conversely, AlphaTuning's non-uniform approach means that for a $b$-bit quantization, there are $b$ individual quantization scales for each weight matrix. Despite having multiple scales, AlphaTuning only fine-tunes one, leaving the rest static. As such, the number of trainable parameters are identical and AlphaTuning seems to offer a larger potential for a well-fitted model, but it can be easily seen that $b - 1$ rest static scales introduced in AlphaTuning have limited usability, and thus the method may be prone to overfitting as evident through empirical results.

In Table 15, we conducted training on GPT-Neo and OPT 1.3B using the Wikitext2 dataset. Interestingly, PEQA, drawing from its methodological advantages, consistently demonstrates superior performance to AlphaTuning by at least 0.7 ppl on the Wikitext2 dataset. Both AlphaTuning and PEQA used channel-wise trainable parameters. Batch size of AlphaTuning is set to 32.

Table 15: The perplexity (PPL) of AlphaTuning and PEQA on Wikitext2 with OPT and GPT-Neo 1.3B. The lower PPL, the better.

| Method | # Bits | OPT 1.3B | GPT-Neo 1.3B |
|---|---|---|---|
| AlphaTuning | 4 | 13.15 | 15.03 |
| PEQA (Ours) | 4 | 12.40 | 14.22 |
| AlphaTuning | 3 | 14.00 | 17.25 |
| PEQA (Ours) | 3 | 13.40 | 15.16 |

Table 16: Learning rate of AlphaTuning in Table 15.

| Method | W Bits | OPT 1.3B | GPT-Neo 1.3B |
|---|---|---|---|
| AlphaTuning | 4 | 1e-4 | 5e-4 |
| AlphaTuning | 3 | 1e-4 | 1e-3 |

## K    Choice of Updating Quantization Scales or Zero-Points

Uniform quantization can represent both asymmetric and symmetric quantizations, hence it's not always necessary to mandate the use of zero-points. This is why adopting a strategy of only learning the scale factor serves as a fundamental and scalable baseline. We opted for this approach to clearly establish its advantages. To determine the efficacy of learning only the scaling factors, we have incorporated additional experiments. By referring to the table below, it's evident that merely optimizing zero-points does not yield effective learning outcomes. Moreover, simultaneously optimizing both zero-points and quantization scales does not present any significant improvement in accuracy either.

Table 17: Perplexity (PPL) of PEQA on the Wikitext2 dataset for LLaMA 7B and LLaMA 13B with weights quantized into 4-bit.

| Method | Zero-points only | Quantization scales only (PEQA) | Both zero-points and quantization scales |
|---|---|---|---|
| LLaMA 7B | 11.56 | 5.84 | 5.86 |
| LLaMA 13B | 9.83 | 5.30 | 5.34 |

## L    Memory Peak on Training

The memory consumption is not solely dictated by the model size but is also influenced by various other factors[6]. Our approach with PEQA inherently offers memory advantages during fine-tuning by striving to minimize both the model size and the number of training parameters. To provide a clear understanding of these benefits, we conducted tests using a single NVIDIA A100-80GB GPU and the causal language modeling code from the HuggingFace repository[7]. Both LoRA and PEQA fine-tuned the LLaMA-7B on the Wikitext2 dataset with a batch size of 2 without gradient accumulation. Our findings indicated that while LoRA peaked at a memory usage of 59GB during optimization, PEQA used just 43GB. Remarkably, this disparity (16GB, 7B) escalates as the model size increases; for instance, a 65B full-precision model under LoRA occupies 130GB, whereas PEQA remarkably uses just 33GB. Additionally, LoRA encountered Out-Of-Memory (OOM) issues at a batch size of 4, whereas PEQA, due to its efficiency, continued training seamlessly.

---

[6]`https://huggingface.co/docs/transformers/perf_train_gpu_one`
[7]`https://github.com/huggingface/transformers/blob/main/examples/pytorch/`
`language-modeling/run_clm_no_trainer.py`

