# OpenReview forum: "Memory-Efficient Fine-Tuning of Compressed Large Language Models via sub-4-bit Integer Quantization"
_NeurIPS.cc/2023/Conference — NeurIPS 2023 poster_

### Official Review · Reviewer_9H2h · 2023-07-06

**Soundness:** 4 excellent
**Presentation:** 4 excellent
**Contribution:** 2 fair
**Rating:** 5
**Confidence:** 3

**Summary:**

This paper presents a method PEQA that combines quantization and parameter-efficient fine-tuning. It takes both advantages, including updating only a tiny fraction of model weights and saving the memory by quantization. The model weights will be decomposed into a matrix of low-bit integers and a scalar vector. During the fine-tuning, only the scalar vector will be updated. The paper has done comprehensive experiments to measure its effectiveness. The proposed method outperforms PTQ and has an acceptable degradation compared to LoRA, but which significantly saved the memory.

**Strengths:**

1. The paper is well-written and easy-to-follow.
2. The idea and the method are clean.
3. The evaluation is comprehensive.

**Weaknesses:**

1. The degradation of the accuracy of the proposed method compared to LoRA is non-negligible.
2. The paper combines two existing techniques rather than inventing a new technique.

**Questions:**

1. Table 2: Why does PEQA have an even lower perplexity in some cases than QAT?
2. How does PEQA compare to the direct combination of LoRA and quantization (e.x. compress to 4-bit, then LoRA on the 4-bit matrices)?
3. What is the intuition behind the decomposition used in PEQA?
4. Table 4: Why PEQA degrades the performance of Five-Shot?

**Limitations:**

1. The degradation of the accuracy of the proposed method compared to LoRA is non-negligible.

---

> ### Author Rebuttal · Authors · 2023-08-09
>
> [**Weakness1**] As the reviewer pointed out, when comparing PEQA with LoRA in terms of the number of trainable parameters (e.g., LLaMA-13B+LoRA vs. LLaMA-13B+PEQA), the degradation of the performance of PEQA relative to LoRA seems to be non-negligible. However, when comparing PEQA with LoRA in terms of the model size (e.g., LLaMA-13B+LoRA vs. LLaMA-30B+PEQA) as shown in Figure 2 (b), PEQA outperforms LoRA in all cases. We believe that it is more fair to compare PEQA with LoRA in terms of the model size than the number of learnable parameters due to the fact that the checkpoint size for trainable parameters is negligible relative to the model size, which makes the overall memory required for deployment totally dominated by the model size.
>
> [**Weakness2**] While the comment is ambiguous as to which two techniques the reviewer thinks our method is based on, we would ike to reiterate that PEQA is a straightforward yet effective way to achieve PEFT and quantization within the same framework.  .PEQA inherits (1) the reduction in the number of trainable parameters and task-switching capability from PEFT and (2) the reduction in the model size and the acceleration of text generation inference from quantization, which makes PEQA more practical and effective than both techniques. Furthermore,, the theoretics and the underlying mechanism that allow PEQA to ourperform previous SOTA  is non-trivial, and we intend to address them in the camera-ready version, if possible.
>
> [**Question1**] In Table 2, as highlighted by the reviewer, PEQA exhibits lower perplexity in certain scenarios compared to QAT. This phenomenon is likely influenced by the straight-through estimator (STE) utilized in the QAT approach we presented. The QAT method in Table 2 is a rudimentary one that simply rounds weights as per Eq. 3 and updates  $W_0$ and $s_0$ using the STE. Notably, the use of STE can sometimes lead to compromised accuracy or increased perplexity due to its approximated gradients possibly failing to converge appropriately. The naive QAT approach we used can experience performance degradation compared to its full-precision counterpart. Our primary intention in comparing PEQA with this basic QAT in Table 2 was to emphasize the rationale for keeping the integer matrix static, rather than to suggest PEQA's superiority over all QAT methods. We will ensure to clarify this perspective in our revisions to avoid any misconceptions.
>
> [**Question2**] The direct combination of LoRA and quantization, such as in the QLoRA approach, is indeed a notable method. As the reviewer highlighted, LoRA can be directly applied to quantized pre-trained models like in the QLoRA technique. However, there are some distinctions and nuances worth mentioning. While QLoRA can indeed achieve accuracy levels comparable to LoRA as detailed in [1], there's an important caveat: it doesn't benefit from the inference acceleration. This is because the weights can't be represented in a quantized format when merging quantized pre-trained weights with LoRA for deployment. Furthermore, if there's a scenario where you wish to fine-tune a deployed quantized PLM for some reason, our methodology might also present itself as an optimal choice. Specifically, if the primary goal is memory-efficient fine-tuning and rapid task-switching of the Quantized PLM, QLoRA could be a viable approach. However, if one's aim is to maintain the quantized form of the PLM and deploy services via memory-efficient fine-tuning with a focus on latency during deployment, PEQA emerges as a potential best-fit.
>
> [**Question3**] The intuition behind the decomposition used in PEQA stems from a desire to inherit from the advantages of both PEFT and quantization. As quantized parameters are expressed as the Hadamard product of full-precision scales and an integer matrix, we freeze the integer matrix while allowing the full-precision scales to be adaptable to a downstream task.
>
> [**Question4**] As highlighted by the reviewer, Table 4 reveals that the five-shot performance of LLaMA-7B + PEQA is inferior to that of LLaMA-7B alone. Conversely, for LLaMA-13B and LLaMA-30B, PEQA yields an improvement in the five-shot performance. In view of these findings, the five-shot performance drop of LLaMA-7B + PEQA can be regarded as a temporary variation, and it would be therefore premature to conclude that PEQA diminishes performance in the five-shot setting.

---

> > ### Comment · Reviewer_9H2h · 2023-08-19
> >
> > Thanks for the clarifications. I have no more question, and would like to keep the original assessment.

---

> > > ### Author Response · Authors · 2023-08-19
> > >
> > > We are pleased that we were able to address your questions. We will certainly incorporate your valuable comments into the revised manuscript.

---

### Official Review · Reviewer_qn8N · 2023-07-06

**Soundness:** 2 fair
**Presentation:** 3 good
**Contribution:** 3 good
**Rating:** 5
**Confidence:** 4

**Summary:**

This paper introduces a new framework, PEQA, for efficiently tuning Large Language Models (LLMs). PEQA uniquely fine-tunes the scale parameters during the instructing tuning phase while keeping the backbone parameters frozen. This approach allows PEQA models to maintain advantages during both the training and inference phases. The primary experiments are performed on the Alpaca datasets. The paper also compares PEQA's performance on various datasets with other efficient tuning methods, such as Lora.

**Strengths:**

The paper is strongly motivated, emphasizing the importance of memory efficiency in tuning methods.

The authors highlight the role of quantization methods in reducing memory usage beyond what is achievable with parameter efficiency methods alone [1].

The paper is well-written and easy to follow.

[1]. https://github.com/huggingface/peft

**Weaknesses:**

Table 1 could be improved by emphasizing that other parameter-efficient tuning methods can also apply post-training quantization methods, such as GPTQ and AWQ.

The baseline should include the post-training quantization methods applied before or after fine-tuning (e.g., Lora and Adapter), and report on the memory usage during inference and training. This is particularly relevant as the HuggingFace PEFT library has incorporated Int8 quantization.

Reference:

[1] QLoRA: Efficient Fine-tuning of Quantized LLMs

**Questions:**

See Weaknesses

**Limitations:**

See Weaknesses

---

> ### Author Rebuttal · Authors · 2023-08-09
>
> [**Weakness1**] We appreciate the reviewer's insightful suggestion. In response, we have updated Table 1 to include both PEFT+PTQ and PTQ+PEFT as per your recommendation. We understand the importance of showcasing that other parameter-efficient tuning methods can also apply post-training quantization methods like GPTQ and AWQ. This modification will be incorporated in our revised manuscript. Thank you for your valuable feedback.
>
> | | Fine-Tuning ||| Deployment ||
> |:----:|:----:|:----:|:----:|:----:|:----:|
> | Method | DRAM | Tuning Speed | DRAM | Inference Speed | Task-Switching |
> | Full Fine-Tuning | $457$GB | Slow | $131$GB | Slow | Slow |
> | PEFT | $131$GB | Fast | $131$GB | Slow | Fast |
> | PEFT+PTQ | $131$GB | Fast | $33$GB | Fast | Slow |
> | PTQ+PEFT | $33$GB | Fast | $33$GB | Slow | Fast |
> | $\textbf{Quantization-Aware PEFT (ours)}$ | $\textbf{33GB}$ | $\textbf{Fast}$ | $\textbf{33GB}$ | $\textbf{Fast}$ | $\textbf{Fast}$ |
>
>
>
> [**Weakness2**] Post-training quantization (PTQ) after fine-tuning has been addressed through LoRA + OPTQ in our work. As for PTQ applied before fine-tuning, it could indeed be conducted as you suggested with methodologies such as QLoRA. As QLoRA is a concurrent work with PEQA, an apple-to-apple comparison between QLoRA and PEQA has not yet been completed due to time constraint of the rebuttal period. However, as can be seen from the updated Table 1, approaches like QLoRA do not ㅈaccelerate inference in the deployment phase.
> QLoRA could indeed be a good method if the requirements solely focus on memory-efficient fine-tuning and quick task-switching of the Quantized PLM. However, if maintaining the quantized PLM form and deploying services through memory-efficient fine-tuning are required, with a particular emphasis on latency during deployment, we suggest that PEQA may be the most suitable choice.
> As for the HuggingFace PEFT library incorporating Int8 quantization, we will study and consider this implementation for our future work and aim to compare it to our method for a more comprehensive evaluation. Thank you for your valuable suggestions.

---

> > ### Author Response · Authors · 2023-08-14
> >
> > Dear reviewer,
> >
> > We genuinely value your feedback and are always open to further discussions.
> >
> > We also would like to highlight the additional implications of memory usage during the fine-tuning process in Table D of uploaded PDF.
> > The memory consumption is not solely dictated by the model size but is also influenced by various other factors[1].
> > Our approach with PEQA inherently offers memory advantages during fine-tuning by striving to minimize both the model size and the number of training parameters.
> > To provide a clear understanding of these benefits, we conducted tests using a single NVIDIA A100-80GB GPU and the causal language modeling code from the HuggingFace repository[2].
> > Both LoRA and PEQA fine-tuned the LLaMA-7B on the Wikitext2 dataset with a batch size of 2 without gradient accumulation.
> > Our findings indicated that while LoRA peaked at a memory usage of 59GB during optimization, PEQA used just 43GB.
> > Remarkably, this disparity (16GB, 7B) escalates as the model size increases; for instance, a 65B full-precision model under LoRA occupies 130GB, whereas PEQA remarkably uses just 33GB.
> > Additionally, LoRA encountered Out-Of-Memory (OOM) issues at a batch size of 4, whereas PEQA, due to its efficiency, continued training seamlessly.
> >
> > Should you have any inquiries or require clarifications about our rebuttal, please don't hesitate to reach out.
> > We are eager to address any concerns and elucidate potential ambiguities in greater depth.
> >
> > [1] https://huggingface.co/docs/transformers/perf_train_gpu_one#anatomy-of-models-memory.
> > [2] https://github.com/huggingface/transformers/blob/main/examples/pytorch/language-modeling/run_clm_no_trainer.py

---

### Official Review · Reviewer_rGqm · 2023-07-07

**Soundness:** 2 fair
**Presentation:** 2 fair
**Contribution:** 2 fair
**Rating:** 5
**Confidence:** 5

**Summary:**

The paper presents a novel approach named Parameter-Efficient and Quantization-aware Adaptation (PEQA) to address the challenges of efficiently fine-tuning and deploying large language models (LLMs). The paper demonstrates PEQA's effectiveness and scalability through extensive experiments, comparing it with competitive baselines across a range of tasks from natural language understanding to generation benchmarks.

**Strengths:**

* The paper centers on the topic of efficient model compression and fine-tuning, framing its approach through a unified method referred to as PEQA.

* The authors' proposed method offers benefits, most noticeably in substantially reducing the memory usage associated with optimizer state saving and the final model size.

* The authors have undertaken a comprehensive evaluation setup which encompasses several benchmark tasks and a variety of model sizes.

**Weaknesses:**

* The overall presentation of the paper could be improved for clarity. Specifically, the meaning of statements such as "Notice that s0 and z0 have nothing to do with a choice of downstream task" on Line 148, and the term "integer quantization indices" on Line 149, are not clear.

* The calculation of memory usage during fine-tuning is not sufficiently explained. While it's understood that the proposed method lowers optimizer state storage by only updating scaling factors, the paper does not address memory usage during gradient calculation, a crucial aspect that often dictates peak memory usage during optimization.

* It's unclear why the choice of only updating scaling factors is advantageous. The paper only compares the proposed method with LoRA, limiting the assessment of its effectiveness. For instance, the BitFit method (Ben Zaken et al., 2022) demonstrates that fine-tuning the LLM can be achieved by only updating biases. Moreover, the paper does not discuss the potential outcomes of optimizing zero-points only or optimizing both zero-points and the scaling factor.

**Questions:**

* The methodology section of the paper leaves some key details unaddressed. Specifically, it's unclear how the gradient of the scaling factors is computed in the proposed model. The paper PACT by Choi, Jungwook, et al. [1] presents two different ways to compute the gradient for the scaling factor/clamp_range for the quantization function. It would be beneficial for the authors to clarify which method was adopted in their work.

[1] Choi, Jungwook, et al. \Pact: Parameterized clipping activation for quantized neural networks.\ arXiv preprint arXiv:1805.06085 (2018).

**Limitations:**

see above.

---

> ### Author Rebuttal · Authors · 2023-08-09
>
> [**Weakness1**] As s0 and z0 are quantization parameters for pre-trained weights W0, our intention was to imply that both s0 and z0 are not related to any downstream task at all. The term “integer quantization index” is also used in [1], which means the rounding of W0/s0, so we refer to  $\overline W_0$  as the integer quantization indices of W0. To enhance clarity, we will revise these expressions in our manuscript.
> [1] https://en.wikipedia.org/wiki/Quantization_(signal_processing)
>
> [**Weakness2**] We appreciate the reviewer's insightful comment regarding the calculation of memory usage during optimization. To clarify, the computation of the gradient for the quantization scale in PEQA necessitates dequantizing the quantized weight to a 16-bit half-precision format. Despite this process, it should be noted that weights from other layers retain their low-bit quantized state. This results in a lesser peak memory usage during optimization for PEQA compared to LoRA. Our proposed model achieves memory efficiency not only during the inference phase but also throughout the optimization process, by minimizing memory consumption related to the model size. We acknowledge the importance of detailing this aspect in our work and will ensure to elaborate on this point in our revision. Thank you for your valuable feedback.
> We also kindly point out that memory usage during fine-tuning is influenced not only by the model size but also by other factors[2]. Hence, in our endeavor to reduce both the model size and the number of training parameters, we adopted PEQA, which provides memory benefits during fine-tuning. To quantify the extent of these memory benefits in actual training scenarios, we measured the memory peaks. Our experimental setup utilized a single NVIDIA A100-80GB GPU and employed the causal language modeling code from the HuggingFace repository [3]. Both LoRA and PEQA fine-tuned the LLaMA-7B on Wikitext2 dataset with a batch size of 2 without gradient accumulation. The results of our tests revealed that LoRA displayed a peak memory usage of 59GB during optimization, whereas PEQA consumed only 43GB. Furthermore, while LoRA experienced Out-Of-Memory (OOM) issues with a batch size of 4, PEQA, owing to its reduced memory usage, was able to proceed with training at this batch size. Accepting the feedback of the reviewer, we will incorporate these findings into our revised manuscript to facilitate a clearer understanding for our readers. Thank you for your valuable feedback.
>
> [2] https://huggingface.co/docs/transformers/perf_train_gpu_one#anatomy-of-models-memory.
> [3] https://github.com/huggingface/transformers/blob/main/examples/pytorch/language-modeling/run_clm_no_trainer.py
>
> [**Weakness3**] Uniform quantization can represent both asymmetric and symmetric quantizations, hence it's not always necessary to mandate the use of a zero-point. This is why adopting a strategy of only learning the scale factor serves as a fundamental and scalable baseline. We opted for this approach to clearly establish its advantages. To determine the efficacy of learning only the scaling factors, we've incorporated additional experiments. By referring to the added table, it's evident that merely optimizing zero-points does not yield effective learning outcomes. Moreover, simultaneously optimizing both zero-points and scaling factors doesn't present any significant improvement in accuracy either. While parameter-efficient adaptation methods, such as BitFit, offer a memory-efficient fine-tuning and preserving quick task-switching abilities, they do not outperform LoRA in terms of accuracy[4]. Our proposed method, PEQA, which updates the uniform quantization scale, was presented as the most straightforward and effective approach that is not only memory-efficient during fine-tuning but also facilitates accelerated inference during deployment, maintains quick task-switching abilities, and delivers acceptable accuracy. We acknowledge this feedback and ensure that the revised manuscript reflects these points accordingly. Given the time constraints of the rebuttal period, our revised manuscript will incorporate additional BitFit experiments to position PEQA alongside other methods as recommended by the reviewers.
>
> | | Zero-point only | Scale-factor only (PEQA) | Both scale-factor and zero-point |
> |:----:|:----:|:----:|:----:|
> | 7B | $11.56$ | $5.84$ | $5.86$ |
> | 13B | $9.83$ | $5.30$ | $5.34$ |.
>
> [4] Hu, Edward J., et al. "Lora: Low-rank adaptation of large language models." The Tenth International Conference on Learning Representations (ICLR), Virtual Event, April 25-29, 2022.
>
>
> [**Question1** ]
> Let the quantization scale $\boldsymbol{s}\_0 = \begin{bmatrix} s_{0, (1)} \newline s_{0, (2)} \newline \vdots \newline s_{0, (n)} \end{bmatrix}$,
> the integer matrix $$ \overline{\boldsymbol{W}}\_0 =
> \begin{bmatrix}
> \overline{W}\_{(1, 1)} , \overline{W}\_{(1, 2)} , \cdots , \overline{W}\_{(1, m)} \newline
> \vdots \newline
> \overline{W}\_{(n, 1)} , \overline{W}\_{(n, 2)} , \cdots , \overline{W}\_{(n, m)}
> \end{bmatrix},$$
> and the quantized weight matrix $$ \widehat{\boldsymbol{W}} =
> \begin{bmatrix}
> \widehat{W}\_{(1, 1)} , \widehat{W}\_{(1, 2)} , \cdots , \widehat{W}\_{(1, m)} \newline
> \vdots \newline
> \widehat{W}\_{(n, 1)} , \widehat{W}\_{(n, 2)} , \cdots , \widehat{W}\_{(n, m)}
> \end{bmatrix}.$$
> Then, the gradient of $\mathcal{L}$ with respect to the scaling factor $\boldsymbol{s}\_0$ is given by
> ${\partial \mathcal{L} \over \partial \boldsymbol{s}\_0} = \begin{bmatrix} {\partial \mathcal{L} \over \partial s_{0, (1)}}  \newline {\partial \mathcal{L} \over \partial s_{0, (2)}}  \newline \vdots \newline {\partial \mathcal{L} \over \partial s_{0, (n)}}  \end{bmatrix}$ where
> ${\partial \mathcal{L} \over \partial s_{0, (i)}} = \Sigma_{j=1}^m \overline{W}\_{(i, j)} {\partial \mathcal{L} \over \partial \widehat{W}\_{(i, j)}}$ for $1 \le i \le n$.

---

> > ### Comment · Reviewer_rGqm · 2023-08-13
> > **Thanks for the response**
> >
> > Thanks for the response.
> > My questions are well addressed. I raise my score accordingly.

---

> > > ### Author Response · Authors · 2023-08-14
> > >
> > > Thank you for your feedback. I'm pleased to hear that your concerns have been addressed. We will make sure to incorporate your valuable suggestions into the revised manuscript.

---

### Official Review · Reviewer_jX2S · 2023-07-11

**Soundness:** 3 good
**Presentation:** 3 good
**Contribution:** 2 fair
**Rating:** 5
**Confidence:** 5

**Summary:**

This paper introduces a model compression method called Parameter-Efficient Quantization-Aware Adaptation (PEQA) to tackle the size challenge of Large Language Models (LLMs) while enhancing their task-specific fine-tuning. The PEQA technique quantizes the fully-connected layers into quantization scales and integer values for initialization, and during downstream tasks, only the quantization scales are fine-tuned, leaving the integer matrix fixed. This approach significantly reduces the parameters required for gradient updates and the memory for loading pre-trained model weights, making fine-tuning more memory-efficient than other techniques such as Post Training Quantization (PTQ) and Quantization-Aware Training (QAT). However, there are several concerns regarding novelty and effectiveness of the proposed methods.

**Strengths:**

- The paper provides an easily understandable explanation of the existing research trends and exhibits good readability.

- By quantizing the pre-trained weights to sub-4bits, the memory cost for LLM fine-tuning has been significantly reduced. This enables a wider range of individuals to fine-tune LLMs according to their desired tasks efficiently. In particular, the existing similar approach, PEFT + PTQ technique, has been studied for models of size smaller than 1.3B. However, PEQA proves to be effective even for models as large as 65B and reduces memory costs.

- Through various experimental results, the paper demonstrates the performance of the PEQA approach in fine-tuning and its application of quantization compared with PTQ and QAT results. This effectively shows how well the method works for the purpose of LLM fine-tuning.

**Weaknesses:**

- This paper's significant contribution, fine-tuning only the quantization scales (termed PEQA), seems similar to AlphaTuning [22]. The paper does not thoroughly discuss the limitations of AlphaTuning, only noting its lack of evaluation on large LLMs.

- The baseline settings in the evaluation setup are unusual. For instance, the authors claim PEQA matches or exceeds QAT's performance. However, QAT's accuracy appears lower than state-of-the-art methodologies: e.g., LLM-QAT[a] demonstrated that 4-bit weight quantization almost reproduces the full-precision accuracy, but QAT in PEQA (Table2, Table3) shows noticeable degradation (LoRA 10.63 vs QAT 11.07). Such understated baselines could potentially mislead readers regarding PEQA's true performance.

[a] LLM-QAT: Data-Free Quantization Aware Training for Large Language Models

- The authors state that PEQA maintains generalization capabilities, yet the results are inconsistent. Table 4 indicates PEQA's zero-shot accuracy matches LoRA, but this improvement disappears in five-shot accuracy. Therefore, it is not clear if PEQA really preserves generalization capability or not.

- There is little understanding of why fine-tuning only the quantization scales is sufficient. Therefore, it is hard to be convinced that the proposed method would work as expected. For example, Table 5 shows that PEQA even noticeably outperforms LoRA (without quantization) LoRA for zero-shot MMLU (and even LoRA+OPTQ outperforms LoRA for LaMMA-7B), which counters the accuracy trends of "LoRA > PEQA > LoRA+OPTQ" consistently shown in all the other tables. It would be desirable to provide insights into such unexpected evaluation results.

- The problem setup in Section 3.1 doesn't align with PEQA's main objectives. The memory-bound issue during text generation inference discussed in this section seems less relevant to PEQA's fine-tuning memory efficiency benefits. Since forward computation isn't performed in the generation style during training, and text generation's memory-bound issue can be resolved by quantization (e.g., LoRA + OPTQ), including this issue in the problem setup may be misleading.

- Comparing 3-bit weight quantization with OPTQ is valuable, but considering OPTQ's known limitations at this level of granularity, a fine-grained quantization comparison might be more convincing. Furthermore, a comparison with recent state-of-the-art weight quantization methodologies, such as LoRA + AWQ, could provide a clearer understanding of PEQA's effectiveness.

**Questions:**

- What could be the reason for PEQA outperforming FP fine-tuning with LoRA (+LoRA) as shown in Table 5? Additionally, the performance of applying OPTQ to LoRA actually increases in LLaMA-7B. Could this be interpreted as the LoRA Fine-Tuning approach not being optimally applied in this experiment?

- From the Five-Shot results in Table 4, we can observe that the performance of PEQA on LLaMA-7B actually decreases, or its fine-tuning effect is insufficient compared to LoRA. Could this result be interpreted as the In-Context Learning ability of LLM not being well-preserved in PEQA Fine-Tuning?

- How does the number of learnable parameters or model size change with a decrease in group size in Table 8? Only perplexity is provided, making it difficult to observe the trade-off together.

**Limitations:**

The authors did not explain the limitations of this paper.

---

> ### Author Rebuttal · Authors · 2023-08-09
>
> [**Weakness1**] When diving deeper into the quantization scale specifics which are learnable parameters of both methods, it's worth noting that PEQA's adherence to uniform quantization means there's only one shared quantization scale for integer weight. Conversely, AlphaTuning's non-uniform approach means that for a $b$-bit quantization, there are $b$ individual quantization scales for each weight matrix. Despite having multiple scales, AlphaTuning only fine-tunes one, leaving the rest static. As such, the number of trainable parameters are identical and AlphaTuning seems to offer a larger potential for a well-fitted model, but it can be easily seen that $b-1$ rest static scales introduced in AlphaTuning have limited utilizability, and thus the method may be prone to overfitting as evident through empirical results. In our appendices, we've showcased a direct performance comparison between the two using a 1.3B model. PEQA, drawing from its methodological advantages, consistently demonstrates a performance that's superior by at least $0.7$ ppl. We also intend to include more detailed studies to address the fundamental differences between previous work and ours.
>
> [**Weakness2**] Although the performance of PEQA can be similar to or better than that of QAT in Table 2, the QAT approach presented in Table 2 is one of the simplest QAT methods that just rounds weights like Eq. 3 and updates $W_0$ and $s_0$ by using the straight-through estimator. As a result, unlike LLM-QAT, the state-of-the-art QAT technique for LLMs (that has been introduced after the NIPs submission deadline), such a rudimentary QAT method in Table 2 can cause performance degradation compared to its full-precision counterpart. As the reviewer mentioned, such a naive QAT approach seems to be an underestimated baseline that might mislead readers to understand that PEQA can either match or surpass QAT methods. However, note that our intention in juxtaposing PEQA with this basic QAT approach in Table 2 was to justify keeping the integer matrix in a frozen state rather than the superiority of PEQA compared to QAT methods. To establish a reliable baseline, we provide full-precision PEFT baseline results, LoRA (higher upper bound), in Table 3.
>
> [**Weakness3 & Question2**] In the case of zero-shot, PEQA's accuracy matches LoRA, while the improvement becomes less pronounced when transitioning to five-shot accuracy. The results in Table 4 & 5 bring forth an interesting phenomenon often termed as the "Alignment tax" [1,2,3]. The concept denotes the potential compromise in performance on traditional NLP tasks as a trade-off for enhanced instruction-following or alignment capabilities. Specifically, as the instruction-following performance heightens, as reflected in Table 5, there's a conceivable penalty that can be observed in tasks like common sense reasoning. This effect, the Alignment tax, is especially more noticeable in smaller models. Piecing everything together, our overarching claim is to emphasize that even after fine-tuning large language models (LLMs) in a compressed state via the PEQA method, the performance in logical reasoning and in-context learning stands its ground.
> [1] Training language models to follow instructions with human feedback.
> [2] A general language assistant as a laboratory for alignment.
> [3] Training a helpful and harmless assistant with reinforcement learning from human feedback.
>
> [**Weakness4 & Question1**] (Please refer to the reply of Weakness 2) We recognize the significance of theoretical aspects of PEQA. We plan to show theoretically why updating only the quantization scales is sufficient as a future work.
> Given that LoRA outperforms LoRA+OPTQ for LLaMA-13B in Table 5, the experimental result where LoRA+OPTQ outperforms LoRA for LLaMA-7B can be regarded as an exceptional occurrence. Yet, PEQA performs better than LoRA for both LLaMA-7B and LLaMA-13B in Table 5, which might be because the number of trainable parameters in PEQA would be more appropriate than that in LoRA (QKVO16) for acquiring the instruction-following ability when fine-tuning a model with an instruction-following dataset (e.g., Alpaca). Similar to the observation that LoRA can be better than full fine-tuning despite possessing fewer learnable parameters, even though the number of learnable parameters in LoRA (QKVO16) is six times more than that in PEQA as seen in Table 7, PEQA would be better suited for developing the instruction-following ability than LoRA.
>
> [**Weakness5**] We kindly remind you that the core objective of PEQA is twofold: achieving memory efficiency during fine-tuning and benefiting from acceleration and quick-task switching during the deployment phase for text generation inference. By employing quantization during fine-tuning, PEQA ensures that weights are preserved in their integer format post fine-tuning. This approach not only tackles the memory constraints during fine-tuning but also addresses the memory-bound issue during text generation inference. As PTQ is applied after the fine-tuning process, PEFT+PTQ doesn't offer the advantages of memory-efficient fine-tuning.
>
> [**Weakness6**] We have performed an additional experiment with LoRA+OPTQ to better elaborate the level of granularity. As pointed out by the reviewer, particularly with LLaMA 13B, we observe that the performance of LoRA+OPTQ improves as the process becomes more fine-grained. However, we continue to see that PEQA consistently demonstrates superior performance when compared to LoRA+OPTQ in Table A(in PDF).
>
> [**Question3**] Parameters in the order of millions have a negligible impact on a model size measured in GBs. When quantizing LLaMA-7B with a group size of 256, a single linear layer has approximately 16 times more learnable scales. When this is scaled linearly, the channel-wise learnable parameter count, which initially was 1.36M, increases by 16 times to reach 21.76M. Yet, this only results in an additional 40MB in memory usage.

---

> > ### Comment · Reviewer_jX2S · 2023-08-19
> > **Thank you for the responses.**
> >
> > I appreciate the authors for the detailed responses and additional experimental results. Although some of the answers do not sufficiently answer the questions, I believe the additional experimental results, as well as the in-depth discussion, would be valuable for future research in this field. Therefore, I raise my original rating.

---

> > > ### Author Response · Authors · 2023-08-19
> > >
> > > We are pleased to note that some of your concerns have been addressed. Your feedback has been instrumental in this process, and we sincerely extend our gratitude for your invaluable insights. Issues that we haven't managed to handle adequately will be designated as future work, underscoring our commitment to continued refinement. We will ensure that your comments are incorporated into the revised manuscript and appreciate the guidance you've provided for our future research endeavors.

---

### Author Rebuttal · Authors · 2023-08-09

Dear Reviewers,

We wish to express our sincere gratitude for your diligent review and insightful feedback on our manuscript.
Your comments have greatly enriched our understanding and enabled us to identify areas where further clarification and improvement were needed.

We have uploaded this summary table in a one-page PDF format in the review system.

Thank you once again for your time and contributions.
We look forward to any further comments or suggestions you may have, and we remain at your disposal for any additional information or clarification.

With kind regards.

---

### Decision · Program_Chairs · 2023-09-21

**Decision:**

Accept (poster)

**Comment:**

This paper proposes a novel quantization-aware PEFT method, Parameter-Efficient and Quantization-aware Adaptation (PEQA), to address the challenges of efficiently fine-tuning and deploying Large Language Models (LLMs). Extensive experiments are conducted on NLU and NLG tasks and various model sizes, compared with with several SOTA baselines. It shows PEQA can effectively save more memory cost than existing PEFT methods.